# What are the impacts of increasing cost-effectiveness Threshold? a protocol on an empirical study based on economic evaluations conducted in Thailand

Wanrudee Isaranuwatchai[1,2]*, Ryota Nakamura[3], Hwee Lin Wee[4], Myka Harun Sarajan[1], Yi Wang[4], Budsadee Soboon[1], Jing Lou[4], Jia Hui Chai[4], Wannisa Theantawee[5,6], Jutatip Laoharuangchaiyot[5,6], Thanakrit Mongkolchaipak[5,6], Thanisa Thathong[5,6], Pritaporn Kingkaew[1], Kriang Tungsanga[6,7], Yot Teerawattananon[1,4]

**1** Health Intervention and Technology Assessment Program (HITAP), Ministry of Public Health, Nonthaburi, Thailand, **2** Institute of Health Policy, Management and Evaluation, University of Toronto, Toronto, Canada, **3** Hitotsubashi Institute for Advanced Study and Graduate School of Economics, Hitotsubashi University, Tokyo, Japan, **4** Saw Swee Hock School of Public Health, National University of Singapore, Singapore, Singapore, **5** Food and Drug Administration (FDA), Ministry of Public Health, Nonthaburi, Thailand, **6** Subcommittee for Development of the National List of Essential Medicines (NLEM), Bangkok, Thailand, **7** Faculty of Medicine, Chulalongkorn University, Bangkok, Thailand

\* wanrudee.i@hitap.net

**Data Availability Statement:** No datasets were generated or analysed during the current study. All

## Abstract

### Background

Economic evaluations have been widely used to inform and guide policy-making process in healthcare resources allocation as a part of an evidence package. An intervention is considered cost-effective if an ICER is less than a cost-effectiveness threshold (CET), where a CET represents the acceptable price for a unit of additional health gain which a decision-maker is willing to pay. There has been discussion to increase a CET in many settings such as the United Kingdom and Thailand. To the best of our knowledge, Thailand is the only country that has an explicit CET and has revised their CET, not once but twice. Hence, the situation in Thailand provides a unique opportunity for evaluating the impact of changing CET on healthcare expenditure and manufacturers' behaviours in the real-world setting. Before we decide whether a CET should be increased, information on what happened after the CET was increased in the past could be informative and helpful.

### Objectives

This study protocol describes a proposed plan to investigate the impact of increased cost-effectiveness threshold using Thailand as a case study. Specifically, we will examine the impact of increasing CET on the drug prices submitted by pharmaceutical companies to the National List of Essential Medicine (NLEM), the decision to include or exclude medications in the NLEM, and the overall budget impact.

relevant data from this study will be made available upon study completion.

**Funding:** This work is funded by the Health Systems Research Institute (HSRI) of Thailand (https://www.hsri.or.th), Grant number HSRI. 64-159. The funders have no contribution to the study such as the study design, data collection, and data analysis.

**Competing interests:** This work is funded by the Health Systems Research Institute (HSRI) of Thailand (https://www.hsri.or.th), Grant number HSRI. 64-159. The funders have no contribution to the study such as the study design, data collection, and data analysis. HITAP is funded by national and international public funding agencies. iDSI is funded by the Bill & Melinda Gates Foundation (OPP1202541), the UK's Department for International Development, and the Rockefeller Foundation. HITAP is also supported by the Access and Delivery Partnership, which is hosted by the United Nations Development Programme and funded by the Government of Japan.

## Materials and designs

Retrospective data analysis of the impact of increased CET on national drug committee decisions in Thailand (an upper middle-income country) will be conducted and included data from various sources such as literature, local organizations (e.g. Thai Food and Drug Administration), and inputs from stakeholder consultation meetings. The outcomes include: (1) drug price submitted by the manufacturers and final drug price included in the NLEM if available; (2) decisions about whether the drug was included in the NLEM for reimbursement; and (3) budget impact. The independent variables include a CET, the variable of interest, which can take values of THB100,000, THB120,000, or THB160,000, and potential confounders such as whether this drug was for a chronic disease, market size, and primary endpoint. We will conduct separate multivariable regression analysis for each outcome specified above.

## Discussion

Understanding the impact of increasing the CET would be helpful in assisting the decision to use and develop an appropriate threshold for one's own setting. Due to the nature of the study design, the findings will be prone to confounding effect and biases; therefore, the analyses will be adjusted for potential confounders and statistical methods will be explored to minimize biases. Knowledge gained from the study will be conveyed to the public through various disseminations such as reports, policy briefs, academic journals, and presentations.

## Introduction

Economic evaluations have been widely used in high- and upper-middle income countries (LMICs) to inform and guide policy-making process in healthcare resources allocation as a part of an evidence package which includes other factors such as equity of access as well as technical and financial feasibility [1]. Economic evaluations compute the incremental cost and benefit of new health interventions compared to the standard of care, in the form of incremental cost-effectiveness ratio (ICER), frequently expressed as incremental cost per quality-adjusted life year (QALY) or incremental cost per disability-adjusted life year (DALY) averted [2–4]. An intervention is considered cost-effective if an ICER is less than a cost-effectiveness threshold (CET), where a CET represents the acceptable price for a unit of additional health gain for which decision-makers, on behalf of the society, are willing to pay [2]. In other words, a CET aims to enable separating interventions that offer good value-for-money from those that do not. This approach allows the same decision making rule applied across different types of health interventions and disease areas [5]. Although there has been a debate around an optimal CET and methodological choices for CET estimation [2,4,6–9], economic evaluation together with either implicit or explicit CETs have commonly been used in price negotiation or value-based pricing around the world [6,10–14].

The use of a CET was first mentioned by United Kingdom's (UK's) National Institute for Health and Care Excellence (NICE) in 1999 [15]. In 2002, the Commission on Macroeconomics and Health launched by the World Health Organization (WHO) presented the concept of linking CET to 1–3 times of gross domestic product (GDP) per capita [16]. Since then, many published cost-effectiveness analyses (CEA) of health interventions in LMICs have explicitly

referred to these WHO criteria as the standards by which each intervention is considered cost-effective [17–20]. However, the use of GDP-guided CETs has been criticised and, recently, WHO recommendations were made not to use it in decision making [21]. Schwarzer et al. [11] conducted a global review and reported in 2015 that only England and Thailand are with explicitly defined CETs whereas Australia, Brazil, Canada, Sweden and the United States of America adopted implicit CETs. Furthermore, a few countries (e.g., Bhutan and Kenya) are working to identify the appropriate CET for their own country context [22].

During the past decades, although there are arguments that the CET is already too high in the UK [23,24], arguments have been made to increase CETs set by NICE [24–27]. The same pressure to increase the CET has also been raised among clinical experts and manufacturers to policymakers in Thailand. For the UK, NICE employs a range of CETs set between £20,000 to £30,000 per QALY, and the CETs have not been changed since the NICE's formation [15]. However, Thailand has revised their CET twice in the past, starting with THB100,000 per QALY in 2008, then increased to THB120,000 per QALY in 2010, and increased again in 2013 to THB160,000 (approximately USD5,000) per QALY. The first CET was set because the policymakers recognized the importance of CET and worked and deliberated with relevant stakeholders to define in 2008. The implementation of the CET led to the discussion that the CET may be too low (which was lower than 1 GDP per capita), and led to further discussion including finally a decision to increase to THB120,000. There was a research study to explore this topic [28], and used to partially to support the increase of CET to THB160,000 in 2013. Since then, there was constant pressure from stakeholders (e.g. industry and professional groups) to increase the threshold again including the recent request in 2019 which led to this study. These Thai CETs have been used to determine if a drug should be listed in the National List of Essential Medicine (NLEM) which is the only pharmaceutical reimbursement list in the country referred by all public health insurance schemes in Thailand NLEM [29]. The same CETs have also been used as benchmarks for listing non-pharmaceutical interventions in the Universal Healthcare Coverage Benefits Package (UCBP) [30]. Increasing CET would theoretically allow more drugs to be included, and subsequently increasing the overall budget impact including potentially influencing access to medicines in the country.

To the best of our knowledge, Thailand is the only country that has explicit CETs and has revised their CET, not once but twice. Further, no empirical study was conducted when the CET was revised in the past. Therefore, the situation in Thailand provides a unique opportunity with empirical dataset for evaluating the impact of changing CET. This information will be vital before deciding whether a CET should be altered in Thailand as well as other setting using CEA information to inform policy decisions. This study protocol describes a proposed plan to investigate the impact of increased CETs using Thailand as a case study. Specifically, we will examine the consequences of increasing CETs on the new medicine prices submitted by pharmaceutical companies to the NLEM, the decision to include or exclude new medications in the NLEM, and the reimbursed medicine budget of the Thai government. Understanding the impact of increasing the threshold would be helpful in assisting the decision to use and develop an appropriate threshold for one's own setting.

## Conceptual framework

The following framework aims to help situate possible scenarios of what could happen if a CET were to be increased. In general, a higher CET may affect the drug prices submitted, the opportunity that drugs will be included in a national list of essential medicine, and budget impact of the reimbursable medicines (Fig 1). There may also be an impact on access to medications and overall population health. Although the long-term impact is outside the scope of

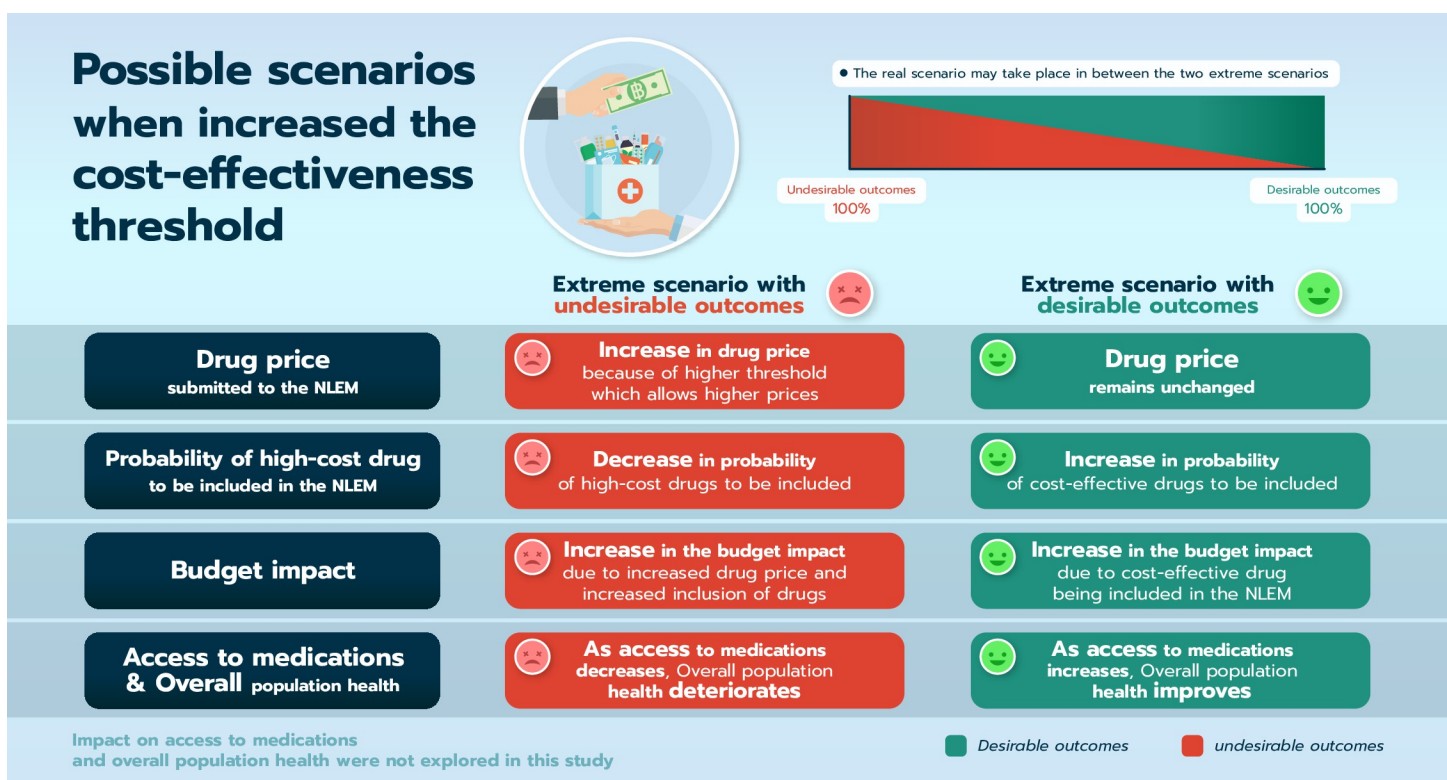

**Fig 1. Framework of possible scenarios when a cost-effectiveness threshold increases.**

this study, these ultimate outcomes (access to medications and overall population health) were added to share a comprehensive view of the framework and were not explored in this study.

Specifically, we considered two extreme scenarios with many possible situations in between. On one end is an extreme scenario with desirable outcomes (ideal scenario) which would imply that everyone acted appropriately and did not take unfair advantage when a CET increases. The other extreme scenario is with undesirable outcomes (problematic scenario) which would happen if relevant stakeholders were to take advantage unfairly when a CET is increased. The real-world outcomes of changing CET could be somewhere between the ideal and the problematic scenarios. Certainly, there are many possible scenarios. The framework is meant to be an overall guide and indicates that the actual scenario will be somewhere in the middle between the two extreme (problematic and ideal) scenarios.

In the ideal scenario with desirable outcomes, a higher CET would not affect the new drug prices submitted by manufacturers because the drug prices will be set based on, for example, the production cost regardless of the national CET. A higher CET may increase both the probability that drugs will be included in the NLEM along with the overall budget impact as more drugs will be considered cost-effective. With these three influences, accessibility to necessary medicines by the needed patients would increase and ultimately improve overall population health.

On the other hand, in the problematic scenario, a higher CET could encourage manufacturers to submit higher new drug prices to NLEM in order to maximise their profits. The probability that drugs will be included in the country's list of essential medicines could increase, remain the same or decrease, depending on the response (prices submitted) from the manufacturers. Nevertheless, the medicine budget would be higher as the CET is higher. Ultimately,

the access to medications and overall population health will be negatively affected due to inappropriate use of limited resources.

## Materials and methods

### Study design and setting

Retrospective data analysis of the impact of increased CET on the national drug committee's decisions in Thailand (an upper middle-income country) will be conducted and will include data from various sources as described below.

### Variables

Fig 2 shows the relationship among dependent variables and independent variable, including variable of interest and confounders, to be used in the analysis.

**Dependent variables.** There are three dependent variables which we will explore: (1) drug price (i.e., we will consider drug price submitted to NLEM by the manufacturers and, if available, final (negotiated) drug price included in the NLEM); (2) decisions (i.e. whether the drug was included in or rejected from the NLEM); and (3) the total (i.e. estimated and actual) budget for each reimbursable drug under the UHC scheme. The variable for drug price will be treated as a continuous variable, and we will conduct two separated models, one for submitted price by the manufacturer, and the other for negotiated price. Decision variable is a categorical variable capturing whether this drug was included in the NLEM. The estimated and actual budget variables are continuous where the estimated budget was the amount of budget impact

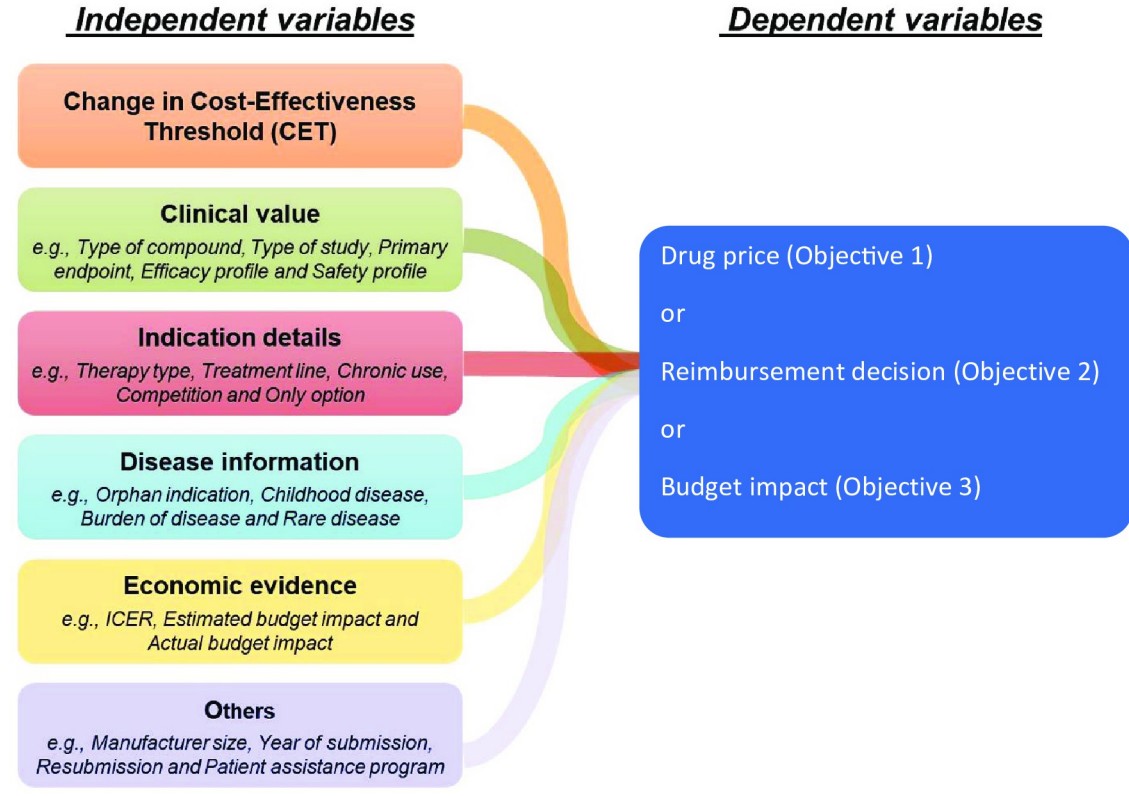

**Fig 2. Independent and dependent variables in the study.**

reported in the study, whereas the actual budget impact was reported from the Price Negotiation Working Group (after negotiation process).

**Independent variable: Variable of interest.**   The independent variable is a CET (categorical variable) which can take the value of THB100,000 (for drug submissions in year 2008–2010), THB120,000 (for submissions in year 2010–2013), or THB160,000 (for drug submissions from 2013 to 2021). As described above, a CET represents the acceptable price for a unit of additional health gain for which decision-makers, on behalf of the society, are willing to pay. From an initial review, there were 48 drug submission over the study period with a total of 267 studies (where one submission may have more than one indication and subsequently more than one study). In 3 periods for CET of THB100,000, THB120,000, and THB160,000, there were 60 studies, 56 studies, and 151 studies, respectively.

**Independent variables: Confounders.**   Potential confounders can be categorized into the following 6 groups: (1) general information; (2) clinical value; (3) indication details; (4) disease information; (5) economic evidence; and (6) others. General information consists of the generic name of the drug submitted to the NLEM, the comparator of the drug in the study, the indication of use of the drug, as well as the year and title of the study. Clinical value includes the following factors: primary endpoint, efficacy profile, safety profile and ICD-10 diagnosis code. Indication details consist of therapy type, treatment line, chronic use, competition, whether the drug is the only option, whether the drug is the only treatment option, and treatment plan (see details in Table 1). Disease information comprises of orphan indication, whether the drug is for children, adults or for all ages, the age of the patient, burden of disease, whether the drug is for a rare disease and if the drug targets a specific sex. ICER, project budget, incremental cost at the governmental and societal perspective, incremental QALYs, estimated budget impact reported in the study and actual budget after price negotiation, actual budget impact, and type of reimbursement are included in economic evidence while others involve type of compound, type of study, manufacturer size, year of submission, submission date, resubmission, number of years after submission, patient assistance program, drug patent, publication, and GDP.

Table 1 summarises the potential confounders identified in literature to be associated with price, reimbursement decision, and budget including their references. These factors will be used in a regression model (e.g., different types of multivariable regression models depending on the outcome). We will apply similar methods to study the impact of an increased CET (as another factor in a regression model) adjusting for these potential confounders on different outcomes of interest (price, reimbursement decision, and budget) based on the objectives.

## Data sources

**Literature review.**   To obtain background information and identify potential factors affecting drug price, reimbursement decisions, and drug budget, we conducted a literature review related to the concept of CET from PubMed and various health technology assessment agency websites such as NICE, the Pharmaceutical Benefits Advisory Committee (PBAC) of Australia, and the Canadian Agency for Drugs and Technologies in Health (CADTH). The following key terms were used: (1) cost-effectiveness threshold; (2) threshold; and (3) willingness-to-pay threshold. We included research articles, academic articles, research reports, and gray literature such as papers presented at academic conferences and articles published on official websites. We restricted the inclusion to English and Thai language studies which were published until September 2021. Studies on the concept of cost-effectiveness threshold or on the effect of setting a cost-effectiveness threshold were included, whereas studies on methodologies used to determine the cost-effectiveness threshold were excluded. Data collected from

**Table 1. Summary of confounders which may associate with price, decision, and budget.**

| Variables | Description | Type of variable | Reference |
|---|---|---|---|
| **CLINICAL VALUE** | | | |
| Efficacy profile | Whether the drug was considered superior (2) /equivalent (1) /inferior (0) compared to comparator | categorical | [31,32] |
| Safety profile | Whether the drug was considered safe (1) or not safe (0) compared to comparator | Binary/ categorical | [31,32] |
| **INDICATION DETAILS** | | | |
| Therapy type | Add-on therapy (0), monotherapy (1), or combination (2) | categorical | [31] |
| Treatment line | First-line treatment (1) or subsequent line (2) | Binary | [31] |
| Chronic use | Whether this drug is for a chronic disease (2) or an acute disease (1) | Binary | [31] |
| Competition | Whether there is more than one manufacturer making this drug: No (0) and Yes (1) | Binary | [31] |
| Only drug option | Whether this drug is the only option for this disease (i.e., there is no other treatment option): No (0) and Yes (1) | Binary | [32–34] |
| Only treatment option | Whether this drug is the only treatment option for this disease (i.e., there is no other treatment): No (0) and Yes (1) | Binary | [32–34] |
| Treatment plan | One time treatment (1), continuous for a period of time (2), or lifetime (3) | Categorical | [31] |
| **DISEASE INFORMATION** | | | |
| Orphan indication | Whether this drug is an orphan drug: No (0) and Yes (1) | Binary | [31,35,36] |
| Childhood disease, adult disease, all-age disease | Whether this submission is for a disease related to childhood disease (0), adult disease (1), or all-age disease (2) | Categorical | [31] |
| Age of the patient | Patient group's age for using the drug | Categorical* | [31] |
| Burden of disease | Estimated burden of disease represented by the number of people affected | Categorical* | [32,35,37] |
| Rare disease | Whether this submission is for a rare disease (1), ultra rare disease (2), or not (0) | Categorical | [31] |
| Disease targeting only certain sex | Whether this disease is only for women (e.g., cervical cancer) (0), men (e.g., prostate cancer) (1), or both (e.g., lung cancer) (2) | Categorical | [31] |
| **ECONOMIC EVIDENCE** | | | |
| ICER | Estimated incremental cost-effectiveness ratio (ICER) | Continuous | [31,34,37] |
| Incremental cost (governmental perspective) | Estimated incremental cost at the governmental perspective | Continuous | [31,34,37] |
| Incremental cost (societal perspective) | Estimated incremental cost at the societal perspective | Continuous | [31,34,37] |
| Incremental QALYs | Incremental Quality-Adjusted Life Year (QALY) | Continuous | [31,34,37] |
| Type of reimbursement | Type of reimbursement (whether it is per case (1) or as a lump sum (0)) | Categorical | [31–33,35,37] |
| **OTHERS** | | | |
| Type of compound | Chemical (1) vs biologic (2) vs biosimilar (3) vs mixed (4) | Categorical | [31] |
| Type of study | Whether the drug's efficacy data are from an RCT: No (0) and Yes (1) | Binary | [31,33,35] |
| Manufacturer size | Manufacturer's company size: big (total revenues of > 1.5 billion) (1) or small (0) | Binary | [31] |
| Year of submission | Year of submission (when the drug was included or rejected) | Continuous | [31] |
| Resubmission | Whether this submission is a resubmission (2) or a submission for the first time (1) | Binary | [31] |
| Patient assistance program | Whether this drug involves a patient assistance program | Binary | [31,35] |
| Gross domestic product (GDP) | GDP at the time of the submission | Categorical | |

Note. "For information" in the reference column refers to how these variables will be used to describe the studies being examined.

* Age of the patient and burden of a disease will be grouped to be a categorical variable; the exact categories will be determined after the review of included studies.

the literature review included the following: (1) title and first author of the study; (2) year of publication; (3) document characteristics; (4) country of the study; (5) importance of a cost-effectiveness threshold; and (6) other relevant details (if any). Findings from the literature review were used to inform potential confounders to include in the analyses, as detailed in Table 1.

**Data sources from local organizations.** Additional information, listed in Table 2, will be requested from various local organizations such as the Thai Food and Drug Administration (FDA), which is a secretary of the national drug committee, as well as the National Health Security Office, which manages the Thai Universal Healthcare Coverage Scheme (UCS), the Social Security Office which manages the Social Security Scheme (SSS) and the Comptroller General's Department which manage the Civil Servant Medical Benefit Scheme (CSMBS). These three public health insurance schemes cover approximately 70 million population of the Thai population and overseas workers in Thailand.

**Stakeholder consultation meetings.** Upon further review of the data obtained from literature review and local organizations, issues caused by lack of evidence will be resolved through consultations with relevant stakeholders and experts in the field such as representatives from the NLEM subcommittee, Health Economic Working Group, and the Universal Coverage Benefit Package (UCBP). We have presented the research protocol at a stakeholder consultation meeting to obtain general feedback on the proposed plan and to initiate collaborations for data access with relevant stakeholders. Once preliminary findings from the project are available, we will present to the stakeholders again for final input. Stakeholders include, but not limited to, representatives from the NLEM subcommittee, pharmaceutical companies, the Development of Rare Disease Care Service, Health Economics Working Groups, the three national health insurance schemes (UCS, SSS, and CSMBS), research communities, and public funding agencies.

## Statistical analysis plan

**Objective 1: Analysis plan for impact of increased CET on drug price.** In Objective 1 (i.e., impact of increased CET on drug prices), we aim to examine the impact of a change in CET on drug prices by using multivariable linear regression models [38,39]. The dependent variable (Y) will be drug price (included in the submission). The variable of interest (X) will be the threshold used (which could be one of the three options: THB100,000, THB120,000, or THB160,000). The regression models will also adjust for potential confounders (Z) as highlighted in the Methods' Variables section. Depending on the final sample size, we may also consider subgroup analysis or interaction terms. For instance, the impact of CET could be different between drugs for rare diseases and drug not for rare diseases. However, such analysis may not be feasible if the sample size is small. The model will also use a cluster robust option to consider the fact that same drugs may be used for different indications and thus have difference prices. The model can be represented by the following function:

**Table 2. Summary of data types and sources.**

| Data Type | Data Sources |
|---|---|
| Median prices of medicines submitted by pharmaceutical companies | Thai Food and Drug Administration (FDA) |
| Outcomes of price negotiation (if any) | Price Negotiation Working Group and the NLEM subcommittee |
| Details of budget impact analysis (e.g., number of patients, total budget, marginal increase in budget) | Universal Coverage Scheme (UCS); Social Security Scheme (SSS); Civil Servant Medical Benefit Scheme (CSMBS) |
| Details of health economic assessments (e.g., ICER) | Health Economic Working Group |
| Details of 10 health economic assessments which were conducted by pharmaceutical companies (e.g., ICER) | Pharamaceutical Research and Manufacturers Association (PReMA) |

*Eq 1 (primary model for Objective 1)*

$$\text{Price}_{ijt} = \alpha_i + \delta_j + \beta * \text{CET}_t + \mathbf{Z_{ijt}}'\gamma + \in_{ijt}$$

(***Price_{ijt}***), the first outcome, will be the submitted drug price of a drug *i* submitted at year *t* by
pharmaceutical company *j*

**$\alpha_i$** = constant term

**$\delta_j$** measures the impact of company *j*

**$\beta$** measures the impact of change in CET on submitted drug prices

**$CET_t$** = Cost effectiveness threshold (THB100,000, THB120,000, or THB160,000 depending on
the submission year)

**$\gamma$** measures the impact of potential confounders (Z) on submitted drug prices

**$Z_{ijt}$** = potential confounders associated with a drug *i* submitted at year *t* by pharmaceutical
company *j* (each confounder will be analysed as individual confounder)

**$t$** = time

In addition to Eq 1, we will conduct a secondary model where drug price will be the final
drug price in the NLEM (after negotiation) if data are available.

**Objective 2: Analysis plan for impact of increased CET on probability of being included
in NLEM.** In Objective 2, we aim to examine the impact of a change in CET on reimburse-
ment decisions by using multivariable logistic regression models [40]. The dependent variable
(Y) will be the decision of the drug being included in the NLEM (yes/no). The variable of inter-
est (X) and other potential predictors (Z) would be similar to those reported in Objective 1.
The model can be represented by the following function:

*Eq 2 (primary model for Objective 2)*

$$logit(p(include\_drug_{ijt} = 1)) = \alpha_i + \beta * \text{CET}_t + \gamma * \text{ICER}_{it} + \mathbf{Z_{it}}'\boldsymbol{\delta}$$

*include_drug_{ijt}* = 1 if *drug_{ijt}*, drug *i* submitted at year *t* by pharmaceutical company *j*, was
included in the NLEM; otherwise, *include_drug_{ijt}* = 0.

**$\alpha_i$** = constant term

**$\beta$** measures the impact of change in CET on the included drug

**$CET_t$** = Cost effectiveness threshold (THB100,000, THB120,000, or THB160,000 depending on
the submission year)

**$\gamma$** measures the impact of ICER on the included drug

**$ICER_{it}$** = Incremental cost-effectiveness ratio of a drug *i* at year *t*

**$Z_{it}$** = potential confounders associated with a drug *i* at year *t*

**$\delta$** measures the impact potential confounders (Z) associated on the included drug

**$t$** = time

**$p$** is probability

**$logit$** is the logit function

**Objective 3: Analysis plan for impact of increased CET on budget impact.** Objective 3 will examine the impact of a change in CET on the estimated and actual budget impact. We will employ the findings from regression models to simulate the change of total budget for all NLEM drugs.

Regression diagnostic tests will be conducted. The above models will be checked for collinearity and homoscedasticity, and robust standard errors will be used. Data will be analysed using STATA, version 16.0 (STATA Corp, College Station, TX, USA) with a statistical significance of $p < 0.05$.

## Discussion

### Potential policy implications

The situation in Thailand where the CET has been increased twice represents a unique case to empirically examine impacts of increasing the CET. Recognising that economic evidence is only part of evidence package in policy making, we believe that better understanding the impact of increased CET on the three different outcomes (i.e., submitted drug prices, decision to include/exclude drugs in the public reimbursement list, and the government budget) could be vital for supporting meaningful deliberation between stakeholders and policymakers on whether the current CET should be revisited in Thailand and other settings in using health economic evaluation to inform policy decisions. While this study will use data from Thailand, the results on the pharmaceutical companies' behavioural responses to a changing CET could be generalizable to other countries. In other words, other countries may expect similar behavioural response from the pharmaceutical companies in response to a changing CET. However, additional consideration on the local healthcare financing setting and healthcare system should be considered.

### Potential limitations and mitigation strategies

This is a study in a single upper-middle income country. Our results may not reflect outcomes that may be found in other settings with different political economy. However, our results on the behavioral responses to the changing CET of the pharmaceutical companies, given their commonalities such as profit driven, can be generalized to other countries, with careful consideration of local healthcare system, local healthcare financing mode, and the existence of other competing pharmaceutical companies.

In Thailand, health economic evaluation is required for only high-cost medicines submitted to NLEM [29]. Also, there are about 50 health economic evaluation studies commissioned for NLEM policy making in the past. These studies reported ICERs of around 80–100 medicines. As such, it needs to be seen whether there are enough data points to make meaningful conclusion of the changing impacts.

Moreover, due to the nature of the study design, the findings are at risk of confounding effect and various types of biases, including misspecification. To account for the confounding effect, relevant confounders were identified by conducting literature review and consulting stakeholders. In hoping to minimize biases, we will conduct applicable regression diagnostics to test for model assumptions (e.g., independence and homoscedasticity) and ensure the model's goodness-of-fit from likelihood ratio test including the Wald test of exogeneity. For endogeneity, multivariable regression analysis can be performed, as well as exploring instrumental variable (IV) methods [41,42]. Since we have developed this study as a policy evaluation of the changes of the CETs, in our models (like many policy evaluations), we treat CET as exogenous treatment variable. We are however aware that one may be concerned if CET is endogenous

since the changes in CET were indeed driven partially by political pressure. Currently, we do not have a list of instrumental variables to address such a policy endogeneity. The use of an IV approach is hence still suggestive and is yet fully committed. However, we would like to retain the possibility of extending our method to address the issue by using an instrumental variable approach.

Furthermore, we will perform a robust regression which can help minimize the mild violation of model assumptions, and robust standard errors will be calculated. The proposed regression analysis is an appropriate way to evaluate the policy change. Other potential approaches could include an interrupted time series analysis (ITS) or regression discontinuity design (RDD), but they were excluded for a combination of reasons. As with ITS, we did not want to aggregate the individual drug level data into a simple time series data. Also, for RDD, with relatively short time period being available in our data, the results of RDD will be very close to event study. Furthermore, RDD is data consuming and usually requires high number of observations. Another example of other study design is to do experts opinion elicitation which is beyond the scope of the current quantitative study.

## Report completion and dissemination of research results

Knowledge gained from the study will be conveyed to the public through various disseminations such as reports, policy briefs, academic journals, and presentations. The research team will share the findings with all relevant stakeholders in both practice and research communities nationally and internationally.

## Acknowledgments

The Health Intervention and Technology Assessment Program (HITAP) is a semi-autonomous research unit in the Ministry of Public Health, Thailand, and supports evidence-informed priority-setting and decision-making for healthcare. We also would like to acknowledge the support from Juthamas Prawjaeng and Phorntida Hadnorntun for the literature review during the early phase of this study.

## Author Contributions

**Conceptualization:** Wanrudee Isaranuwatchai, Kriang Tungsanga, Yot Teerawattananon.

**Data curation:** Myka Harun Sarajan, Budsadee Soboon, Jing Lou, Jia Hui Chai, Wannisa Theantawee, Jutatip Laoharuangchaiyot, Thanakrit Mongkolchaipak, Thanisa Thathong.

**Funding acquisition:** Wanrudee Isaranuwatchai, Kriang Tungsanga, Yot Teerawattananon.

**Methodology:** Wanrudee Isaranuwatchai, Ryota Nakamura, Hwee Lin Wee, Yi Wang, Kriang Tungsanga, Yot Teerawattananon.

**Project administration:** Budsadee Soboon, Thanisa Thathong.

**Supervision:** Kriang Tungsanga, Yot Teerawattananon.

**Writing – original draft:** Wanrudee Isaranuwatchai, Ryota Nakamura, Myka Harun Sarajan, Yi Wang, Kriang Tungsanga, Yot Teerawattananon.

**Writing – review & editing:** Wanrudee Isaranuwatchai, Ryota Nakamura, Hwee Lin Wee, Yi Wang, Budsadee Soboon, Jing Lou, Jia Hui Chai, Wannisa Theantawee, Jutatip Laoharuangchaiyot, Thanakrit Mongkolchaipak, Thanisa Thathong, Pritaporn Kingkaew, Kriang Tungsanga, Yot Teerawattananon.

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
