## [Decision Letter · Decision Letter 0]

18 May 2022

PONE-D-22-01139What are the Impacts of Increasing Cost-effectiveness Threshold? A Protocol on an Empirical Study based on Economic Evaluations conducted in ThailandPLOS ONE

Dear Dr. Isaranuwatchai,

Thank you for submitting your manuscript to PLOS ONE. After careful consideration, we feel that it has merit but does not fully meet PLOS ONE’s publication criteria as it currently stands. Therefore, we invite you to submit a revised version of the manuscript that addresses the points raised during the review process. While reviewer 1 is more prone to recommend only minor revision, reviewer two asked for a deeper improvement of the paper. After going through the points raised by reviewer 2, I recommend the authors implement the suggested improvements whenever possible. If the authors refrain from implementing some suggestions, please explain why you decided not to implement such changes. 

We look forward to receiving your revised manuscript.

Kind regards,

Jaume Garcia-Segarra

Academic Editor

PLOS ONE

Journal Requirements:

“iDSI is funded by the Bill & Melinda Gates Foundation (OPP1202541), the UK’s Department for International Development, and the Rockefeller Foundation. HITAP is also supported by the Access and Delivery Partnership, which is hosted by the United Nations Development Programme and funded by the Government of Japan.”

 “This work is funded by the Health Systems Research Institute (HSRI) of Thailand (https://www.hsri.or.th), Grant number HSRI. 64-159. The funders have no contribution to the study such as the study design, data collection, and data analysis.”

3. Please upload a new copy of Figure 2 as the detail is not clear. Please follow the link for more information: https://blogs.plos.org/plos/2019/06/looking-good-tips-for-creating-your-plos-figures-graphics/

Reviewers' comments:

Reviewer's Responses to Questions

**Comments to the Author**

1. Does the manuscript provide a valid rationale for the proposed study, with clearly identified and justified research questions?

Reviewer #1: Yes

Reviewer #2: Partly

2. Is the protocol technically sound and planned in a manner that will lead to a meaningful outcome and allow testing the stated hypotheses?

Reviewer #1: Yes

Reviewer #2: Partly

3. Is the methodology feasible and described in sufficient detail to allow the work to be replicable?

Reviewer #1: Yes

Reviewer #2: Yes

4. Have the authors described where all data underlying the findings will be made available when the study is complete?

Reviewer #1: Yes

Reviewer #2: Yes

5. Is the manuscript presented in an intelligible fashion and written in standard English?

Reviewer #1: Yes

Reviewer #2: Yes

6. Review Comments to the Author

You may also provide optional suggestions and comments to authors that they might find helpful in planning their study.

Reviewer #1: This study protocol entitles “What are the impacts of increasing cost-effectiveness threshold? A protocol on and empirical study based on economic evaluations conducted in Thailand.” Overall, the protocol is interesting and the future findings of the study would be of interest and benefit to Thailand and other countries. Several comments are as follows:

1. Authors should provide reasons why the CET of Thailand has been revised twice within 5-6 years (2008-2013) and constant up until present (2014-2022)? What were the driven factors to specify the amount of CET of Thailand?

2. Based on the conceptual framework (Figure 1), three variables, drug price, probability of high-cost drug, budget impact, will be measured following to the objectives of the study. How do the authors plan to measure the access to medications & overall population health? Should it be one of main objective similar to the other 3 objectives?

3. For the extreme scenario with undesirable outcomes from the conceptual framework, the budget impact might not necessarily increase when an access to medication decreases? BIA is the product of cost and population. Although drug price increases, the population uptake reduces, it is possible to have same/lower/higher budget.

4. Table 1 shows summary of confounders. Do authors consider GDP/GNI or country income as the confounder? It should impact on CET based on WHO guideline of cost-effectiveness.

5. Equation 1 and equation 2, the term “Z” refers to all 6 groups of potential confounders?

6. The process of NLEM consideration might take time which would be different in each country. The CET was revised twice during the year of 2008-2013. The drug was submitted at one year, like 2008, but study was done and considered into NLEM at another year, like 2010. Which year will be used for analysis?

Reviewer #2: This research protocol addressed an interesting issue on the effect of increasing CET, using Thailand as a case study. Nevertheless, there are several comments and issues that need further clarification, as follows:

1. Authors specified 3 independents variables (Y) on page 5, however, it was not clear stated nor consistently stated throughout the manuscript.

a. Y = drug price, authors mentioned that it is the price submitted to NLEM and price after negotiation if available. Could authors further clarify if the regression model for Price will be 2 separated models (i.e., one for submitted price, the other for negotiated price) or both prices will be used in 1 model (if available). In my view, the price after negotiation might be more relevant.

b. Y = total budget (i.e., actual and estimated budget) for each reimbursed medicine (page 5). However, on page 11, authors mentioned that Y was the change in total budget from estimated to actual budget. Please clarify whether Y is the change between actual and estimated budget? Or else? Please also be consistent in several places (i.e. abstract, objective, and method)

2. As this is a protocol manuscript and confounder is an important issue, although authors mentioned that the list of confounders was derived from literature review authors should clearly explain the rationale for each confounder in more details instead of just adding the references in table 1. Furthermore, some confounders seemed not to derived from literature review and no clear rationale was provided. For example,

a. General information

b. budget of the project

c. submission DATE

d. ICD-10

e. Whether drug target different sex

f. Patient’s age group

g. Etc.

3. For table 1, authors mentioned that it was the summary of confounders. However, some do not seem to be confounders but dependent variables or independent variables. Please check carefully. In addition, some variables in table 1 might not meet criteria of being confounders such as general information and might not need to be in the model. In addition, each regression model might require different set of confounders. So, it might be good to provide potential list of confounders separately for each regression model.

4. It was also unclear how authors incorporated the confounders (mentioned in table 1) into the regression model. Please further specify type of variables along with its level if they are categorical variables. For example, ICER, ICD-10, age of the patient, year of submission, etc.

a. ICER, there might be more than 1 ICER for 1 medicine in a single medicine. Which ICER will be put in the model and it will be put as continuous variable or categorical (e.g. higher or lower than CET). Please further classify.

5. Regarding clinical value, how authors obtained the clinical value? Whether it will be derived from 1 RCT or from meta-analysis? Or else? In addition, comparator and primary endpoint along with their clinical value might differ across different studies. So, which study will the author used to derive clinical value should be clearly mentioned.

6. Whether there should be other confounders? For example, whether price in other settings and level of innovative should be considered as confounders for the model on “price”?

7. Please also provide a list of potential instrumental variables that might be included in the analysis along with its rationale.

8. Table 2, authors mentioned that other relevant details will be derived from PREMA. It might be interesting to learn if PREMA will have all the rest of information? If not, what are the other sources of information?

9. Authors should provide more detail information on how many decisions were made during the CET of 100,000, 120,000, and 160,000 Baht so the reader could have some idea on the sample size/data points that will be available in the analysis.

10. Besides the method used in this study (regression model using data from 1 country, where CET was increased twice), whether there are other methods/study design that could provide answer on the impact of increasing CET on price, budget, or decision? If so, why the authors should this method over other methods?

7. PLOS authors have the option to publish the peer review history of their article (what does this mean?). If published, this will include your full peer review and any attached files.

Reviewer #1: No

Reviewer #2: No

---

## [Author Response · Author response to Decision Letter 0]

6 Jul 2022

Response to Reviewer: Manuscript ID: PONE-D-22-01139 

What are the Impacts of Increasing Cost-effectiveness Threshold? A Protocol on an Empirical Study based on Economic Evaluations conducted in Thailand

PLOS ONE

Independent Review Report, Reviewer 1

EVALUATION

This study protocol entitles “What are the impacts of increasing cost-effectiveness threshold? A protocol on and empirical study based on economic evaluations conducted in Thailand.” Overall, the protocol is interesting, and the future findings of the study would be of interest and benefit to Thailand and other countries. 

RESPONSE: Thank you.

Several comments are as follows:

1) Authors should provide reasons why the CET of Thailand has been revised twice within 5-6 years (2008-2013) and constant up until present (2014-2022)? What were the driven factors to specify the amount of CET of Thailand?

RESPONSE: This topic has an interesting history. The policymakers in Thailand recognized the importance of CET and worked and deliberated with relevant stakeholders to define the first CET in 2008. After the implementation, the policymakers realized that the original CET of THB100,000 was lower than 1 GDP per capita, and may be too low so there were discussion and finally decisions to increase to THB120,000. There was a research study to explore this topic, and used to partially to support the increase of CET to THB160,000 in 2013. Since then, there was constant pressure from stakeholders (e.g. industry and professional groups) to increase the threshold again including the recent request in 2019 which led to this study.

2) Based on the conceptual framework (Figure 1), three variables, drug price, probability of high-cost drug, budget impact, will be measured following to the objectives of the study. How do the authors plan to measure the access to medications & overall population health? Should it be one of main objective similar to the other 3 objectives?

RESPONSE: From the conceptual framework, access to medications and overall population health were mentioned as examples of long-term outcomes. Although the long-term impact is outside the scope of this study, these ultimate outcomes (access to medications and overall population health) were added to share a comprehensive view of the framework. We have highlighted in the manuscript in the “Conceptual framework” section on page 4. 

3) For the extreme scenario with undesirable outcomes from the conceptual framework, the budget impact might not necessarily increase when an access to medication decreases? BIA is the product of cost and population. Although drug price increases, the population uptake reduces, it is possible to have same/lower/higher budget.

RESPONSE: Certainly, there are many possible scenarios. The framework is meant to be an overall guide and indicates that the actual scenario (impact on budget for example) will be somewhere in the middle of the gradient. We have elaborated this point in the manuscript in the “Conceptual framework” section on page 4.

4) Table 1 shows summary of confounders. Do authors consider GDP/GNI or country income as the confounder? It should impact on CET based on WHO guideline of cost-effectiveness.

RESPONSE: Recent literature has now indicated that WHO is moving away now from suggesting CET to be 1-3 GDP per QALY. Furthermore, Thailand’s CET was not based only on GDP. Therefore, this variable was not included in the model. Additionally, GDP is measured yearly, and, given the data period, there may be multicollinearity between CET and GDP. 

With that being said, we recognized the reviewer’s point and a related observation that countries may be able to have higher CET and give more funding to healthcare system when they become richer (i.e. higher GDP). Therefore, we have included this variable in our analyses and have revised the manuscript accordingly.

5) Equation 1 and equation 2, the term “Z” refers to all 6 groups of potential confounders?

RESPONSE: Yes, Z is a vector representing all confounders in the equation. In the model when we do analysis, we will include each confounder in the model. We have clarified that in the manuscript. 

6) The process of NLEM consideration might take time which would be different in each country. The CET was revised twice during the year of 2008-2013. The drug was submitted at one year, like 2008, but study was done and considered into NLEM at another year, like 2010. Which year will be used for analysis?

RESPONSE: The analysis will follow the year that the drug was considered into NLEM because the CET used will depend on when study of the corresponding drug was considered by NLEM subcommittee. We have clarified this point in the Methods section of the revised manuscript. 

 

Independent Review Report, Reviewer 2

EVALUATION

This research protocol addressed an interesting issue on the effect of increasing CET, using Thailand as a case study.

RESPONSE: Thank you. 

Nevertheless, there are several comments and issues that need further clarification, as follows:

1) Authors specified 3 independents variables (Y) on page 5, however, it was not clear stated nor consistently stated throughout the manuscript.

a. Y = drug price, authors mentioned that it is the price submitted to NLEM and price after negotiation if available. Could authors further clarify if the regression model for Price will be 2 separated models (i.e., one for submitted price, the other for negotiated price) or both prices will be used in 1 model (if available). In my view, the price after negotiation might be more relevant.

RESPONSE: We agree with the reviewer that the price after negotiation (if negotiation occurs) would be interesting to the real-world setting as well. Our plan is to conduct 2 separated models, one for submitted price, and the other for negotiated price, as now highlighted in the manuscript. 

b. Y = total budget (i.e., actual and estimated budget) for each reimbursed medicine (page 5). However, on page 11, authors mentioned that Y was the change in total budget from estimated to actual budget. Please clarify whether Y is the change between actual and estimated budget? Or else? Please also be consistent in several places (i.e. abstract, objective, and method)

RESPONSE: Total budget would refer to the actual budget. We have revised all relevant parts and ensure consistency throughout the paper. 

2) As this is a protocol manuscript and confounder is an important issue, although authors mentioned that the list of confounders was derived from literature review authors should clearly explain the rationale for each confounder in more details instead of just adding the references in table 1. Furthermore, some confounders seemed not to derived from literature review and no clear rationale was provided. For example,

a. General information

b. budget of the project

c. submission DATE

d. ICD-10

e. Whether drug target different sex

f. Patient’s age group

g. Etc.

RESPONSE: We have added more clarification for each confounder. Thank you.

3) For table 1, authors mentioned that it was the summary of confounders. However, some do not seem to be confounders but dependent variables or independent variables. Please check carefully. In addition, some variables in table 1 might not meet criteria of being confounders such as general information and might not need to be in the model. In addition, each regression model might require different set of confounders. So, it might be good to provide potential list of confounders separately for each regression model.

RESPONSE: Details on dependent and independent variables are provided in their own section. We have revised Table 1 to focus mainly on the confounders. Currently, we aim to include all confounders in the initial regression model and will work towards creating a parsimonious model.

4) It was also unclear how authors incorporated the confounders (mentioned in table 1) into the regression model. Please further specify type of variables along with its level if they are categorical variables. For example, ICER, ICD-10, age of the patient, year of submission, etc.

RESPONSE: We have updated Table 1 to include the type of variable accordingly. 

a. ICER, there might be more than 1 ICER for 1 medicine in a single medicine. Which ICER will be put in the model and it will be put as continuous variable or categorical (e.g. higher or lower than CET). Please further classify.

RESPONSE: ICER will be put in the model as a continuous variable. One medicine may have more than one ICER if the medicine is meant for more than one indication. We will include an ICER value per indication. 

5) Regarding clinical value, how authors obtained the clinical value? Whether it will be derived from 1 RCT or from meta-analysis? Or else? In addition, comparator and primary endpoint along with their clinical value might differ across different studies. So, which study will the author used to derive clinical value should be clearly mentioned.

RESPONSE: We will be using the clinical value used in each HTA study which follow the Thai National Methodological HTA guideline . The guideline suggests the use of systematic review and meta-analysis of RCTs as the first option, and, without systematic review and meta-analysis, a single RCT is preferable followed by cohort study, case control, and observation study. Expert opinion would be the last sort of clinical outcomes mentioned in the guidelines.

6) Whether there should be other confounders? For example, whether price in other settings and level of innovative should be considered as confounders for the model on “price”?

RESPONSE: We thank the reviewer for the suggestion. We have tried to identify as many confounders as we can while taking into account issue on data availability. We will explore the possibility of obtaining data on price in other settings as another confounder while also exploring further with the policy-makers on whether this information affected the decision-making process in Thailand. We realize that we will not be able to control for all potential confounders and will highlight this limitation (based on the nature of the study design) in the discussion.

7) Please also provide a list of potential instrumental variables that might be included in the analysis along with its rationale.

RESPONSE: Since we have developed this study as a policy evaluation of the changes of the CETs, in our models, like most policy evaluations, we treat CET as exogenous treatment variable. We are however aware that one may be concerned if CET is endogenous, since the changes in CET were indeed driven partially by political pressure. Currently we do not have a list of instrumental variables to address such a policy endogeneity. The use of an IV approach is hence still suggestive and is yet fully committed. However, we would like to retain the possibility of extending our method to address the issue by using an instrumental variable approach. In the revised manuscript, we have deleted the sentence concerning an IV approach from Method section, and have discussed about the potential extension in Discussion section. 

8) Table 2, authors mentioned that other relevant details will be derived from PREMA. It might be interesting to learn if PREMA will have all the rest of information? If not, what are the other sources of information?

RESPONSE: From initial consultation with the Thai FDA, there were a small number of HTA studies (N=10) where the final reports had to be obtained from PREMA. We have revised the manuscript to reflect this new information.

9) Authors should provide more detail information on how many decisions were made during the CET of 100,000, 120,000, and 160,000 Baht so the reader could have some idea on the sample size/data points that will be available in the analysis.

RESPONSE: We have updated the manuscript to add more details on number of decisions made during each CET period. 

10) Besides the method used in this study (regression model using data from 1 country, where CET was increased twice), whether there are other methods/study design that could provide answer on the impact of increasing CET on price, budget, or decision? If so, why the authors should this method over other methods?

RESPONSE: We used real-word data to examine the impact of changing CET. In order for this method to work, we need a setting with explicit CET and CET had to change over time. These conditions make Thailand a unique country to conduct such study.

Given the policy change in Thailand which can serve as a natural experiment, we believe that our proposed regression analysis is an appropriate way to evaluate the policy change. Other potential approaches could include an interrupted time series analysis (ITS) or regression discontinuity design (RDD). We have excluded these approaches for a combination of reasons. As with ITS, we did not want to aggregate the individual drug level data into a simple time series data. Also, for RDD, with relatively short time period being available in our data, the results of RDD will be very close to event study. Furthermore, RDD is data consuming and usually requires high number of observations. Another example of other study design is to do experts opinion elicitation which is beyond the scope of the current quantitative study.

---

## [Decision Letter · Decision Letter 1]

4 Aug 2022

PONE-D-22-01139R1What are the Impacts of Increasing Cost-effectiveness Threshold? A Protocol on an Empirical Study based on Economic Evaluations conducted in ThailandPLOS ONE

Dear Dr. Wanrudee Isaranuwatchai,

Thank you for submitting your manuscript to PLOS ONE. After careful consideration, we feel that it has merit but does not fully meet PLOS ONE’s publication criteria as it currently stands. Therefore, we invite you to submit a revised version of the manuscript that addresses the points raised during the review process.

We received two reports, and both reviewers are satisfied with your improvements after your revision. So consider this decision as a conditional acceptance once you address the final points raised by reviewer 2.

We look forward to receiving your revised manuscript.

Kind regards,

Jaume Garcia-Segarra

Academic Editor

PLOS ONE

Journal Requirements:

Reviewers' comments:

Reviewer's Responses to Questions

**Comments to the Author**

1. Does the manuscript provide a valid rationale for the proposed study, with clearly identified and justified research questions?

Reviewer #1: Yes

Reviewer #2: Yes

2. Is the protocol technically sound and planned in a manner that will lead to a meaningful outcome and allow testing the stated hypotheses?

Reviewer #1: Yes

Reviewer #2: Partly

3. Is the methodology feasible and described in sufficient detail to allow the work to be replicable?

Reviewer #1: Yes

Reviewer #2: Yes

4. Have the authors described where all data underlying the findings will be made available when the study is complete?

Reviewer #1: Yes

Reviewer #2: No

5. Is the manuscript presented in an intelligible fashion and written in standard English?

Reviewer #1: Yes

Reviewer #2: Yes

6. Review Comments to the Author

You may also provide optional suggestions and comments to authors that they might find helpful in planning their study.

Reviewer #1: The authors have addressed the comments made satisfactorily and have made changes to the manuscript accordingly. No further comments.

Reviewer #2: The revised version is much clearer. However, there are few points to clarify and consider.

1. In the introduction part, please consider add the recent study, which found that the empirical CET might be less than 1 GDP per capita (see, Gloria MAJ, Thavorncharoensap M, Chaikledkaew U, Youngkong S, Thakkinstian A, Culyer AJ. A Systematic Review of Demand-Side Methods of Estimating the Societal Monetary Value of Health Gain. Value Health. 2021 Oct;24(10):1423-1434)

2. Impact on population health were not included in the analysis so please note in the conceptual framework figure.

3. Table 1:

• Variables that in the reference columns filled with “for information” are not confounders. Please take them out from table 1 as table 1 provides the list of confounders.

• It seemed that if the type of variable was mentioned as “text” (e.g. primary endpoint, ICD), it will not be in the regression model. If so, they are not confounders as well. If they are not confounders please take them out from table 1.

• Age of the patient – if it is the group, it should not be continuous variable

• Burden of disease- should it be categorical variable instead of continuous variable (e.g., < xxx, xxx-xxx, etc.)

• For variable “publication”, any publication will be counted or only effectiveness, cost-effectiveness?

4. Table 2: Details of 10 health economic assessment … should added “ that were conducted by pharmaceutical company”

5. Page 11: Under equation 2. For description of ICER it , Z ij, whether the word “included” should be taken out. Please change Zij to Zit

7. PLOS authors have the option to publish the peer review history of their article (what does this mean?). If published, this will include your full peer review and any attached files.

Reviewer #1: No

Reviewer #2: No

---

## [Author Response · Author response to Decision Letter 1]

5 Sep 2022

Response to Reviewer: Manuscript ID: PONE-D-22-01139R1 

What are the Impacts of Increasing Cost-effectiveness Threshold? A Protocol on an Empirical Study based on Economic Evaluations conducted in Thailand

Independent Review Report, Reviewer 1

EVALUATION

Reviewer #1: The authors have addressed the comments made satisfactorily and have made changes to the manuscript accordingly. No further comments.

RESPONSE: Thank you.

Independent Review Report, Reviewer 2

EVALUATION

Reviewer #2: The revised version is much clearer. However, there are few points to clarify and consider.

1. In the introduction part, please consider add the recent study, which found that the empirical CET might be less than 1 GDP per capita (see, Gloria MAJ, Thavorncharoensap M, ChaikledkaewU, Youngkong S, Thakkinstian A, Culyer AJ. A Systematic Review of Demand-Side Methods of Estimating the Societal Monetary Value of Health Gain. Value Health. 2021 Oct;24(10):1423-1434)

RESPONSE: We added the highlighted reference accordingly. Thank you. 

2. Impact on population health were not included in the analysis so please note in the conceptual framework figure.

RESPONSE: We added in the Figure that impact on population health was not included in the analysis. 

3. Table 1:

• Variables that in the reference columns filled with “for information” are not confounders. Please take them out from table 1 as table 1 provides the list of confounders.

RESPONSE: Noted and we revised Table 1 accordingly. 

• It seemed that if the type of variable was mentioned as “text” (e.g. primary endpoint, ICD), it will not be in the regression model. If so, they are not confounders as well. If they are not confounders please take them out from table 1.

RESPONSE: Noted and we revised Table 1 accordingly. 

• Age of the patient – if it is the group, it should not be continuous variable

RESPONSE: We revised the variable type with explanation that age of the patient will be grouped to be a categorical variable; the exact categories will be determined after the review of included studies. We will know more once we review the studies. 

• Burden of disease- should it be categorical variable instead of continuous variable (e.g., < xxx,xxx-xxx, etc.)

RESPONSE: We revised accordingly. Thank you. 

• For variable “publication”, any publication will be counted or only effectiveness, cost-effectiveness?

RESPONSE: Only the cost-effectiveness studies will be counted. 

4. Table 2: Details of 10 health economic assessment … should added “ that were conducted by pharmaceutical company”

RESPONSE: We revised the table accordingly. Thank you. 

5. Page 11: Under equation 2. For description of ICER it , Z ij, whether the word “included” should be taken out. Please change Zij to Zit

RESPONSE: We removed the word “included” for both ICER it and Z it.

---

## [Editor Report · Decision Letter 2]

8 Sep 2022

What are the Impacts of Increasing Cost-effectiveness Threshold? A Protocol on an Empirical Study based on Economic Evaluations conducted in Thailand

PONE-D-22-01139R2

Dear Dr. Wanrudee Isaranuwatchai,

We’re pleased to inform you that your manuscript has been judged scientifically suitable for publication and will be formally accepted for publication once it meets all outstanding technical requirements.

Kind regards,

Jaume Garcia-Segarra

Academic Editor

PLOS ONE
---

## [Editor Report · Acceptance letter]

23 Sep 2022

PONE-D-22-01139R2 

What are the Impacts of Increasing Cost-effectiveness Threshold? A Protocol on an Empirical Study based on Economic Evaluations conducted in Thailand 

Dear Dr. Isaranuwatchai:

I'm pleased to inform you that your manuscript has been deemed suitable for publication in PLOS ONE. Congratulations! Your manuscript is now with our production department. 

Kind regards, 

on behalf of

Dr. Jaume Garcia-Segarra 

Academic Editor

PLOS ONE